# The Hidden Danger of Unintentional Child Injuries in an Urban Domestic Environment: Considering Unintentional Injuries from Another Angle

**DOI:** 10.3390/ijerph22071068

**Published:** 2025-07-03

**Authors:** Ping Tang, Qin Fan, Jingmin Sun, Jianlin Ji, Liling Yang, Wenjuan Tang, Qunfeng Lu

**Affiliations:** 1Nursing Department of Children’s Hospital, School of Medicine, Shanghai Jiao Tong University, Shanghai Children’s Hospital, Shanghai 200062, China; tangping@shchildren.com.cn; 2General Surgery Department of Children’s Hospital, School of Medicine, Shanghai Jiao Tong University, Shanghai Children’s Hospital, Shanghai 200062, China; fanq@shchildren.com.cn; 3Outpatient Department/Emergency Department of Children’s Hospital, School of Medicine, Shanghai Jiao Tong University, Shanghai Children’s Hospital, Shanghai 200062, China; sunjm@shchildren.com.cn; 4School of Nursing, School of Medicine, Shanghai Jiao Tong University, Shanghai 200025, China; jijianlin@sjtu.edu.cn; 5Nursing Department of the Sixth People’s Hospital, School of Medicine, Shanghai Jiao Tong University, Shanghai 200233, China; youngll955@163.com

**Keywords:** unintentional injury, children, home environment, safety hazards, China

## Abstract

Background: Unintentional injuries are the primary cause of death and disability among children. This study aimed to examine the current status of home environments for children aged 0–6 years in the Shanghai area of China and assess the factors that pose safety hazards for unintentional injuries within households. Methods: A cross-sectional survey was carried out in Shanghai between November 2021 and October 2023. Results: Parents from 1825 families, with 929 (50.90%) boys and 896 (49.10%) girls, participated in this research. In all, 752 children (41.21%) experienced unintentional injuries. The home environment posed a high risk of unintentional injuries in 1008 families (55.23%), medium risk in 381 families (20.88%), and low risk in 436 families (23.89%). The results showed a negative correlation between the occurrence of unintentional injuries and the status of the family environment; children in families with high-risk home environments were more likely to experience unintentional injuries (odds ratio [OR] = 1.490, confidence interval [CI] = 1.216–1.826), fall injuries (OR = 1.605, CI = 1.268–2.031), and external injuries (OR = 1.578, CI = 1.159–2.148). Conclusions: Parents should enhance their safety awareness by focusing on potential hazards at home and taking appropriate measures to improve the home environment, thereby creating a safe and comfortable setting for the healthy growth of their children.

## 1. Introduction

Unintentional injuries are the primary cause of death and disability among children; globally, over 270,000 infants and children under the age of 5 years die each year as a result of injuries [1]. Injury is defined as bodily harm caused by the acute transfer of mechanical, thermal, electrical, chemical, or radiation energy or by sudden deprivation of heat or oxygen [2]. Unintentional injuries refer to injuries that occur without obvious premeditation, including motor vehicle accidents, choking, drowning, poisoning, burns, falls, and injuries associated with sports and recreational activities [2]. A significant proportion of deaths caused by unintentional injuries occur in developing countries, where the mortality rate of children related to injuries is four to six times higher than that in developed countries [3], making unintentional injuries a major public health issue worldwide [4].

Existing systematic reviews state that the determinants of unintentional injuries in children are influenced by various factors, such as neighbourhood, family, parents, and children [5,6]. Although child-related factors are the strongest risk factors for injuries, family factors should not be overlooked because a chaotic family environment increases the likelihood of unintentional injuries [5]. Furthermore, the younger the child, the greater the role of the family in the occurrence of unintentional injuries [5]. Previous studies have demonstrated that the home, as the primary place of activity for infants and toddlers, has been identified as the main location for such injuries, accounting for 60% of unintentional injuries in children under 1 year of age [7,8,9]. Factors such as home safety conditions, housing structures, types of building materials, and the presence of safety facilities are closely related to the occurrence of unintentional injuries in children [10]. Children aged 0–6 years are in the infant and toddler stages, characterised by a strong curiosity about the outside world. This period is crucial for understanding the external environment; however, children at this stage do not develop adequate danger recognition or self-protection abilities, making them highly susceptible to unintentional injuries [11,12].

Despite the high incidence of unintentional injuries among children, they can be prevented and controlled. In 2015, the Royal Society for the Prevention of Accidents in England estimated that unintentional injuries accounted for 60–65% of preventable deaths among children aged under 5 years [13]. Unintentional injuries occurring in the domestic environment represent a significant contributor to preventable deaths and serious disabilities, particularly among preschool-aged children, and are attributed to the increased time spent at home and the various hazards present in these environments.

As a coastal city in China, which is a developing country, Shanghai has transformed into one of the most advanced first-tier cities in the nation after 45 years of reform and opening. Urbanisation has brought about various challenges, including a dense population, ageing, poorly maintained buildings, and an increasing number of high-rise structures, which increase the risk of unintentional injuries to children at home. Despite the numerous studies on unintentional injuries in children, research on the risk of such injuries in home settings is lacking. Understanding the current living conditions of children at home is essential for formulating effective measures to prevent unintentional injuries. Thus, we formulate the research hypothesis that a hazardous family environment may partially indicate the likelihood of unintentional injuries in children.

We conducted a cross-sectional study to examine the home environments of children aged 0–6 years in urban Shanghai and analysed the potential risk factors contributing to unintentional injuries. We hope to provide valuable insights for the development of relevant intervention measures and policies in the future.

## 2. Materials and Methods

### 2.1. Study Design and Participants

This cross-sectional study was conducted in Shanghai from November 2021 to October 2023 and involved parents of preschool children from nine kindergartens and childcare centres.

### 2.2. Participants Slection

By cluster random sampling method, parents from one kindergarten in each of the seven administrative districts of Huangpu, Xuhui, Changning, Jing’an, Putuo, Hongkou, and Yangpu in downtown Shanghai were selected. The sample size calculation used a multiplier of 5 to 20 times the number of items (54), resulting in a required sample size of 270 to 1080.

### 2.3. Criteria for Inclusion

Inclusion criteria were as follows: (1) parents are local residents of Shanghai or have lived in Shanghai for a long time, (2) their children are aged between 3 and 6 years, attending preschool, (3) their children are enrolled in kindergartens in the Shanghai area, (4) the children have no difficulties in reading and writing or cognitive impairments, and (5) voluntary participation. Before the study, informed consent was obtained from the participants with the support of kindergarten teachers; all included parents had signed consent forms.

### 2.4. Research Tools

#### 2.4.1. Demographic Questionnaire

The research team designed a demographic questionnaire that gathered general information about the children, including sex, age, number and types of injuries, presence of underlying diseases, and family information, such as housing nature and type, floor materials, primary caregiver’s educational level, and annual family income.

#### 2.4.2. Environment Scale of Unintentional Injury

Home environment status was assessed using the Unk ‘Environment Scale of Unintentional Injury’ developed by Wang et al. [10]. This scale aims to identify risk factors within home environments that are likely to cause unintentional injuries in children. The final version of the scale encompassed six dimensions: falls, external force injuries, burns/scalds, poisoning, foreign body injuries, and animal injuries, with 15, 12, 12, 7, 6, and 2 items, respectively, totalling 54 items. A 5-point Likert scale was used to score each item from 1 to 5. The overall Cronbach’s α coefficient was 0.87, while the Cronbach’s α coefficients for each dimension ranged from 0.50 to 0.70 [10]. We obtained authorisation to use the scale from the original authors via email before commencing the study.

### 2.5. Data Collection

The principal investigator contacted each childcare centre to establish a cooperative relationship. A survey team was subsequently formed, comprising members of the research team and teachers from childcare centres. The research leader provided standardised training to the survey team members on the questionnaire collection procedures. All questionnaires were printed, bound to booklets, and sent to each childcare centre by courier. During the drop-off and pick-up periods, the teachers conducted ‘one-on-one’ surveys with parents who met the inclusion criteria and provided informed consent. Participants completed the questionnaire anonymously and independently; the survey took 10–15 min. Questionnaires were collected on the spot and checked for quality; any missing content was promptly supplemented.

### 2.6. Statistical Analysis

All questionnaires were double-entered into the Epidata software, verified for accuracy, and imported into IBM SPSS Statistics for Windows, version 24.0. Descriptive statistics were expressed as mean ± standard deviation (SD) and percentage. Spearman’s correlation analysis was performed. Odds ratios (ORs) and 95% confidence intervals (CIs) were used to examine associations between two variables. Statistical significance was set at *p* < 0.05.

### 2.7. Ethical

This study complies with the Helsinki Declaration. Informed consent was obtained from participants prior to data collection, and the study was approved by the Ethics Committee of Shanghai Children’s Hospital on 11 September 2023, with the following approval number: 2023R086-E01. The scales used in this study were authorised in writing by the authors of the scales, and permission was also obtained from the institutions where the data were collected.

## 3. Results

### 3.1. General Information

A total of 1874 questionnaires were distributed, and 1825 valid questionnaires were returned, yielding an effective response rate of 97.39% (Table 1).

### 3.2. Incidence of Unintentional Injuries in Children

Overall, 752 children (41.21%) experienced unintentional injuries, and the distribution of specific injury types demonstrated a logical hierarchy. Falls constituted the primary injury category, affecting 723 children (39.62%) and accounting for the vast majority of cases alone. This was followed by external force injuries, affecting 418 children (22.90%) and highlighting a significant gap between the two most common injury types.

The less frequent but notable categories of burns or scalds affected 92 children (5.04%), whereas poisoning incidents affected 34 children (1.86%). Foreign body and animal injuries occurred in 44 (2.41%) and 56 (3.06%) children, respectively. This sequential presentation, from the most prevalent falls to relatively rare animal injuries, reveals a logical gradient of injury occurrence (Table 2).

### 3.3. Scores on the Home Unintentional Injury Environment Scale

The overall score on the Home Unintentional Injury Environment Scale was 94.86 ± 28.80. The home environment posed a high risk of unintentional injuries in 1008 households (55.23%), medium risk in 381 households (20.88%), and low risk in 436 households (23.89%). Over half of the households were categorised as having a high-risk environment for unintentional injuries (Table 3).

### 3.4. Analysis of Risk Factors for Unintentional Injuries in the Home

Spearman’s correlation analysis was performed to examine the relationship between the occurrence of unintentional injuries among children and scores on the Home Unintentional Injury Environment Scale. The results indicated a negative correlation between the occurrence of unintentional injuries and the scores across each dimension of the scale. The results of the binary logistic regression showed that high-risk environmental conditions were associated with unintentional injuries (OR = 1.490), with significant associations noted for falls (OR = 1.605) and external injuries (OR = 1.578). For some injury categories (e.g., poisoning), the number of cases was very low, resulting in odds ratios with wide confidence intervals and non-significant *p*-values (Table 4).

## 4. Discussion

We surveyed 1825 parents of preschool children living in a metropolitan area of a developing country. The results indicated a relatively balanced sex ratio among the participants, with both public and private kindergartens included in the study. Information on primary caregivers revealed that mothers played a crucial role in childcare, followed by grandparents. This means in first-tier cities, such as Shanghai, China, despite the fast-paced lifestyle and high work pressure, mothers remain committed to devoting more time to their preschool children. Conversely, fathers’ participation in childcare requires further improvement. The surveyed parents had a high level of education, with more than half having received a college education, and the families were economically well off and belonged to China’s middle class.

In recent years, increased attention from all sectors of society and ongoing public education by professionals have contributed to the declining trend in medical visits for unintentional injuries [14]. However, unintentional injuries among children remain a global public health issue that requires continuous attention [4]. In this study, despite more than half of the surveyed parents having a relatively high level of education and access to knowledge on preventing unintentional injuries, the occurrence of such injuries remains concerning. The study results indicated that among the 1852 children surveyed, (752 41.21%) experienced unintentional injuries. This finding corroborates that the age range of 0–6 years is the peak period for unintentional injuries among children, consistent with other scholars’ findings [9,15]. Unintentional injuries include falls, external force injuries, burns/scalds, poisoning, foreign object injuries, and animal injuries. Falls are the most common unintentional injury among children. This may be because kindergarten children are in the stage of learning to walk and are unsteady, while slightly older children are active and playful, and thus are more susceptible to falls during activities such as chasing and roughhousing [16]. In addition, external force injuries and burns/scalds have high incidence rates after falls. These injuries increase the number of emergency visits and may threaten the safety of children in severe cases. These findings suggest that parents should be vigilant while supervising their children at home to prevent unintentional injuries.

The situation regarding unintentional injury within households was not optimistic. The mean total score of the Household Unintentional Injury Environment Scale in our study was 94.86 ± 28.80, suggesting a high risk. More than half (55.23%) of the households exhibited a high risk of unintentional injuries within their homes. This finding reiterates that the home is the location with the highest incidence of unintentional injuries among children, with nearly half of the injury-related deaths among children aged under 5 years occurring at home [17,18]. The home is the primary place for children to live and play; however, an unsafe home environment poses significant risks to their healthy development [19]. Therefore, scholars have proposed a “5E” comprehensive intervention to address unintentional injuries among children, including educational, engineering, enforcement, economic, and emergency care and first aid interventions [20]. Although recent research has focused on enhancing parents’ emergency response capabilities concerning preschool children’s unintentional injuries [21,22], this study indicates that prevention requires improving parents’ safety awareness and treatment skills, as well as enhancing the home environment to mitigate risk factors, such as installing window guards, softening floors, and ensuring that household appliances are not damaged.

With the popularisation of health science in China, parents’ ability to recognise and prevent unintentional injuries among their children has significantly improved [23]. However, the results of this study indicate that many parents do not pay sufficient attention to optimising their home’s physical environment. As the home is the primary setting for unintentional injuries among children, unintentional injuries among children need to be re-evaluated from the perspective of the home environment. Regarding fall injuries, while most falls are not serious, some can result in severe injury, disability, or even fatal outcomes. Falls are a leading cause of fatal and serious head injuries among young children [24]. Numerous studies have indicated that falling is the most common cause of unintentional injury among children in China [9,25]. This may be attributed to home decoration preferences in China, where easily cleanable materials such as tiles and wooden floors are favoured, and few families carpet the entire floor. These harder types of flooring increase the likelihood of falls requiring medical attention among children. Concerning injuries caused by external forces, many parents have not implemented adequate preventive measures against external injuries, leaving sharp furniture corners exposed in children’s activity areas and increasing the risk of injury. Regarding burn injuries, the results indicate that Chinese families are not adequately equipped with fire-safety devices. A recent survey of safety hazards in Chinese households revealed that 64% of the families did not install gas alarms [26]. While children may suffer minor burns or scalds from hot water or appliances, home fires pose a direct threat to their safety. Thus, a smoke alarm and fire extinguisher must be present in every home. Accidental poisoning is a major cause of unintentional injuries among children and the leading cause of PICU admission [27]. The incidence of poisoning among patients under 20 years of age is 1484 per 100,000 [28], with young children being the primary victims [28,29,30]. Previous research in China has indicated that pesticides were the main toxins, followed by drugs [31]. However, recent studies have found that drugs have become the main toxins involved in child poisoning cases in China (43.5%) [29]. In India, another developing country, the incidence of pesticide poisoning (20.6%) slightly exceeds that of drug poisoning (19.2%) [32]. The incidence of drug poisoning is expected to increase with socioeconomic development, making it imperative to strengthen prevention and control efforts. In home life, it is crucial for parents to increase awareness of the prevention of child poisoning and to effectively manage oral medications, insecticides, cleaning products, and cosmetics to reduce the incidence. Regarding foreign body injuries, aspiration and ingestion of foreign bodies were the most common injuries among children under 3 years of age, as they often explore the world by chewing [33]. Foreign body inhalation is a life-threatening emergency that requires prompt identification and treatment to prevent fatal complications [34]. Most ingested foreign objects pass through the gastrointestinal tract without complications. However, some foreign objects may cause problems if they remain in the tract for a long time [35]. Recently, many children have been reported to undergo surgery for having ingested or inhaled items, including non-corrosive objects such as toy parts and peanuts, as well as hazardous items such as magnetic beads and batteries that cause gastrointestinal issues [35]. Parents can mitigate the risk of foreign object injuries in living areas by enhancing supervision and learning the Heimlich manoeuvre, which can save a child’s life in the case of choking [35]. Last but not least, this research found that animal injuries affected 56 children (3.06%), ranking it as one of the less common but still significant causes of unintentional injuries among the surveyed population. In comparison, a cross-sectional study conducted by Shalini et al. across paediatric injury studies reported that animal-related injuries accounted for an average of 3.0% of all unintentional childhood injuries, indicating that our findings fall within the expected range [36]. Therefore, households with children should implement safety protocols, such as supervised interactions between children and pets [37]. Training children on proper pet handling and educating parents on immediate first-aid procedures for animal-inflicted injuries, as outlined in the World Health Organization’s guidelines on childhood injury prevention, can further reduce the risk of such incidents. [38].

This study has several strengths, including data collection from kindergartens rather than hospital emergency departments, enhanced representativeness, and a relatively large sample size covering nearly the entire urban area of Shanghai. However, this study has certain limitations. First, the use of a cross-sectional survey method and the absence of multivariate analysis to account for potential confounders limited the depth of data analysis. Second, the sample selection did not strictly adhere to random sampling, and this could have affected the results. Third, preschool children with intellectual disabilities and other abnormalities were excluded, resulting in a sample that may not be highly representative. Furthermore, the confirmatory factor analysis results of the Home unintentional Injury Environment Scale used in this study are open to question (the RMSEA, CFI, and GFI values were 0.07, 0.61, and 0.71, respectively); some items have a standard deviation greater than 2 or less than 0.5. These results may suggest duplication or issues with item interpretation, or identical responses from most participants, which hinders their ability to discriminate. Moreover, this study only used items to represent and analyse potential risk factors for unintentional injuries, without conducting an in-depth analysis of the influencing factors. Finally, because all results were derived from cross-sectional data, the conclusions should be interpreted with caution.

## 5. Conclusions

The issue of children’s unintentional injuries requires ongoing attention as the state of the household environment remains a matter of concern. Nearly half of the children surveyed experienced unintentional injuries. More than half of the households reported having home environments conducive to unintentional injuries. It is recommended that parents enhance their ability to prevent accidental injuries, schools implement educational courses on accident prevention, and the government enacts relevant laws and regulations, thereby creating a safe and comfortable space for children’s healthy growth.

## Figures and Tables

**Table 1 ijerph-22-01068-t001:** General demographic data and basic family information for 1825 children.

Variable	Total (n = 1825)
Age	4.63 ± 0.87
Gender/n (%)
Boy	929 (50.90)
Girl	896 (49.10)
Nature of kindergarten/n (%)
Public kindergarten	1210 (66.30)
Private kindergarten	615 (33.70)
Primary caregiver/n (%)
Mother	1503 (82.36)
Father	21 (1.15)
Grandmother	285 (15.61)
Grandfather	9 (0.54)
Others	7 (0.38)
Caregiver literacy level/n (%)
Primary School or Below	89 (4.88)
Middle school	195 (10.68)
High School	361 (19.78)
University/college	1018 (55.78)
Postgraduate and above	162 (8.88)
Annual household income/n (%)
<80,000 RMB	54 (2.96)
80,000 to 150,000 RMB	183 (10.03)
150,000 to 300,000 RMB	563 (30.85)
300,000 to 500,000 RMB	624 (34.19)
500~1 million RMB	335 (18.36)
>1 million RMB	66 (3.62)
Type of the house/n (%)
Rental housing	277 (15.18)
Own housing	1538 (84.27)
Others	10 (0.55)
Building Type/n (%)
Villadom	33 (1.81)
One-storey house	21 (1.15)
Multistorey building	1771 (97.04)

**Table 2 ijerph-22-01068-t002:** Unintentional injuries of the tested children.

Types of Unintentional Injuries	N (%)
Fall injury	723 (39.62)
External injury	418 (22.90)
Burns/scalds	92 (5.04)
Poisoning	34 (1.86)
Foreign body injury	44 (2.41)
Animal injury	56 (3.06)
Total	752 (41.21)

**Table 3 ijerph-22-01068-t003:** Scores on the environment scale of unintentional injury.

Items/Mean ± Std Deviation	N	Mean	Std. Deviation
1. The bathroom is covered with non-slip MATS.	1820	2.32	1.78
2. Windows and balconies are equipped with protective Windows or guardrail, and meet safety requirements: vertical arrangement, children cannot turn over, and you cannot drill into the gap.	1820	1.91	1.49
3. The ground is flat without protrusions or tilts, such as thresholds and sliding doors.	1821	1.37	0.82
4. The carpeted floor is neat, without curling or loose.	1821	2.73	2.25
5. The stairs have handrails, and meet the safety requirements: there is a vertical arrangement, children cannot climb over, and you cannot drill through the gap.	1821	2.51	2.16
6. The slope of the stairs is moderate, and children can walk without difficulty.	1821	2.41	2.11
7. Stair steps are not damaged.	1821	2.36	2.09
8. The surface of the stairs has good friction, or a step anti-slip strip is installed.	1821	2.86	2.14
9. The indoor stairs are equipped with protective measures that children cannot pass through, such as protective fences.	1821	3.66	2.36
10. The access to the roof is protected and children cannot pass through.	1821	2.67	2.20
11. Bay windowsill height is moderate, and children cannot climb it.	1821	2.35	1.94
12. Windows and balconies are not placed under the furniture or items that can be climbed, such as chairs, tables, etc.	1821	1.64	1.15
13. Clean floor, with no oil, water or other liquids.	1821	1.20	0.52
14. Clean ground, with no wires, ropes, children’s toys or other obstacles.	1821	1.39	0.72
15. The corridor and home lighting are in good condition.	1821	1.19	0.53
16. There are no plants with thorns or sharp leaves in the children’s usual range of activities, such as cacti, aloe vera, etc.	1821	1.19	0.66
17. Windows, mirrors and other large glass products are intact.	1821	1.11	0.48
18. Children’s toys are without damage or sharp parts, such as darts, small shovels or other metal toys.	1821	1.29	0.63
19. There are no sharp objects, such as knives, razors, scissors, needles, fishhooks, forks, chopsticks, etc., found in the children’s usual range of activities.	1819	1.32	0.65
20. There are no fragile items, such as vases, glasses, perfume bottles, glass ornaments, etc., in the children’s regular range of activities.	1820	1.41	0.73
21. There are no sharp, hard protrusions on the walls accessible to children, such as nails, LCD TV stands, etc.	1820	1.33	0.69
22. The sharp parts of the furniture accessible to children are covered with leather covers, sponges and other protective appliances, such as coffee table corners, table corners, bed corners, etc.	1820	1.58	0.95
23. The items installed on the ceiling are fixed well, such as lamps, ceiling fans, etc.	1820	1.14	0.46
24. The items hanging on the wall are fixed well, such as picture frames, wall lamps, wall fans, air conditioners, etc.	1820	1.17	0.49
25. The items on the furniture should be properly placed and not easily touched by children, such as books, decorations, etc.	1820	1.49	0.78
26. Doors and drawers are equipped with anti-extrusion protection devices.	1820	2.12	1.40
27. There are no high-speed rotating items in the children’s usual range of activities, such as running fans and washing machines.	1819	1.53	0.98
28. There is a working smoke alarm in the home.	1820	3.67	2.08
29. A working fire extinguisher is installed in the home and stored in the correct location (easily accessible and not prone to fire).	1820	3.66	2.06
30. The water dispenser used is designed with child safety protection.	1820	2.82	2.22
31. The hot water pipe at home is fully wrapped without exposure.	1820	1.45	1.23
32. Pressure cookers and rice cookers have no ageing or damage during their service life.	1820	1.18	0.65
33. There is no food with a high temperature within the children’s normal range of activities, such as hot rice, vegetables, soup, water, etc.	1820	1.28	0.62
34. There are no containers or electrical appliances with a high temperature within the children’s usual activity range, such as hot water bottles, hot kettles, stoves, electric ovens, electric heaters, electric irons, etc.	1820	1.28	0.62
35. Non-combustible items such as matches, lighters, candles, paraffin, gas lamps, mosquito-repellent incense, fireworks, etc., are usually in the children’s area.	1820	1.26	0.65
36. No unextinguished cigarette butts are left in the ashtray at home.	1820	1.83	1.76
37. Electricshock-prone facilities such as power sockets and lighting switch buttons are installed with protective devices or are not within the reach of children.	1820	1.48	0.84
38. Electrical appliances are kept away from water areas such as tubs and shower heads.	1820	1.17	0.50
39. The circuit is installed in the wall, and the external line insulation layer is intact without exposure.	1820	1.13	0.44
40. The gas stove or natural stove in the home is equipped with a protection device or a ventilation hole at the switch knob.	1820	1.28	0.77
41. The gas water heater is not installed in the bedroom or bathroom.	1820	1.20	0.75
42. At home, there are emetic drugs suitable for children, such as ipecac syrup.	1820	3.34	2.00
43. Non-alcoholic beverages, such as beer, liquor, wine, etc., are commonly used in children’s range of activities.	1820	1.37	0.87
44. There are no household necessities, such as washing powder, toilet cleaner, insecticides, pesticides, etc., in the children’s regular activities.	1820	1.43	0.84
45. There are no medicines in the area of children’s regular activities, or the packaging has a child-safe cover design.	1820	1.38	0.78
46. No cosmetics or self-care products, such as facial cleanser, toner, cream, body lotion, toilet water, etc., are in the children’s regular activities.	1820	1.63	1.00
47. Children’s toys are of moderate size, cannot be put directly into the mouth, and have no fragile or easily removable parts.	1780	1.56	0.83
48. There are no round, hard, small, sharp objects, such as coins, buttons, etc., in the children’s usual range of activities.	1780	1.55	0.84
49. No round, hard, small, sharp food, such as hard candy, nuts, raisins, peas, fish bones, bones, etc., in children’s usual range of activities.	1780	1.56	0.85
50. Water storage containers are in a safe state within the children’s usual activity range, such as with a lid or without water storage. (Water storage containers include reservoirs, buckets, water tanks, fish tanks, bathtubs, toilets, etc.)	1780	1.32	0.77
51. There are no string-like items that are easy to wrap around children in their normal range of activities, such as twine, ribbon, thick thread, curtain rope, rubber bands, yo-yos, necklaces, clothes with ropes, etc.	1780	1.39	0.71
52. There are no plastic film products, such as plastic bags, plastic wrap, balloons or balloon pieces, etc., in the children’s regular activities.	1780	1.47	0.80
53. No cats, dogs or other pets are kept in the home.	1779	1.52	1.30
54. There are no animals or insects such as rats, cockroaches or spiders in the home.	1779	1.45	0.90
Dimension/± S
Total score of fall injury dimension	1820	32.57	15.43
Total score of external injury dimension	1818	16.69	5.43
Total score of burn/scald dimension	1820	22.20	7.51
Total score of poisoning injury dimension	1820	11.64	4.17
Total score of foreign body injury dimension	1780	8.84	3.65
Total score of animal injury dimension	1779	2.97	1.77
Risk grouping/n (%)
High-risk group	1008 (55.23)
Medium-risk group	381 (20.88)
Low-risk group	436 (23.89)

**Table 4 ijerph-22-01068-t004:** Scores on the home environment scale.

Variable	Total (n = 1825)	r	OR	95%CI
Fall injury	32.57 ± 15.43	−0.103 *	1.605	1.268–2.031
External injury	16.69 ± 5.43	−0.131 *	1.578	1.159–2.148
Scald	22.20 ± 7.51	−0.141 *	1.192	0.737–1.928
Poisoning	11.63 ± 4.17	−0.157 *	1.600	0.145–17.677
Foreign body injury	8.84 ± 3.65	−0.158 *	1.955	0.807–4.738
Animal injury	2.97 ± 1.77	−0.087 *	1.735	0.914–3.296
Total	94.86 ± 28.80	−0.153 *	1.490	1.216–1.826

* indicates *p* < 0.01.

## Data Availability

The data are not publicly available because they contain information that could compromise research participant privacy and consent, but are available from the corresponding author upon reasonable request.

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
