# Peer review of "The Hidden Danger of Unintentional Child Injuries in an Urban Domestic Environment: Considering Unintentional Injuries from Another Angle"

_ijerph, 2025, doi:10.3390/ijerph22071068_

Round 1
Reviewer 1 Report
Comments and Suggestions for Authors
The article presents a solid structure and sound methodological framework, offering results that are highly relevant for the prevention of unintentional childhood injuries in urban environments. The large sample size, the use of a validated assessment tool, and the analytical approach contribute to the scientific value of the study.
However, in order to enhance the manuscript’s suitability for publication, several improvements are recommended. First and foremost, a professional language revision is essential to correct the numerous grammatical and syntactic issues that currently affect the clarity and fluency of the text. Additionally, the manuscript would benefit from condensing redundant sections, particularly in the Methods and Discussion, to make the content more concise and reader-friendly.
From a presentation standpoint, the inclusion of summarizing tables or graphical representations—such as visual summaries of the environmental risk scores or injury frequencies—could significantly enhance the accessibility of the data and support the interpretation of key findings.
Lastly, the Discussion section would be strengthened by a deeper reflection on the practical implications of the study’s results. This includes more explicit references to how the findings could inform public health strategies, policy-making, or the design of targeted interventions aimed at improving home safety for children. By translating the evidence into concrete recommendations, the manuscript can make a more meaningful contribution to the field.

The manuscript would benefit from a thorough professional English language revision. While the overall meaning is generally understandable, the text contains numerous grammatical errors, awkward sentence structures, and inconsistent terminology. These issues affect the clarity and fluency of the manuscript, particularly in the Introduction and Discussion sections. A careful review by a native English speaker or professional editing service is strongly recommended to ensure the language meets the standards required for international scientific publication.
Author Response
Dear professor:
I am writing to express my sincere gratitude for the invaluable feedback provided by you on my manuscript titled “The Hidden Danger of Unintentional Child Injuries in Urban Domestic Environment:Considering Unintentional Injuries from Another Angle .” I appreciate the time and effort you have dedicated to evaluating my work, and your insightful comments have significantly enhanced the quality of my submission.
In response to your suggestions, I have made several revisions to improve the clarity and rigor of my manuscript. Specifically, I have addressed the following points:
Comments1:The section contains numerous syntactic and grammatical errors (e.g., “had been identified the main location” should be corrected to “has been identified as the main location”). A thorough language revision is strongly recommended
Response1: Thank you for pointing out the grammatical deficiencies in this manuscript.In response to this issue, we have made comprehensive revisions to the manuscript.
Comments2:The use of expressions such as “lead to energy exceeding the human threshold” is ambiguous. A clearer definition of the concept of “unintentional injury” is suggested.
Response2:Agree. We have revised the relevant definition of unintentional injury to “Injury is defined as bodily harm caused by the acute transfer of mechanical, thermal, electrical, chemical, or radiation energy, or by sudden deprivation of heat or oxygen. Unintentional injuries refer to types of harm that occur without obvious premeditation, including motor vehicle accidents, choking, drowning, poisoning, burns, falls, and injuries associated with sports and recreational activities.”
Comments3:Greater consistency in terminology is recommended; the alternation between “unintentional injuries” and “accidental injuries” is not explained.
Response3:Agree. Key definition has been unified, with all occurrences of "accidental injuries" in the manuscript replaced by "unintentional injuries."
Comments4:The study objective is presented at the end of the section but should be more clearly and concisely highlighted at the conclusion of the introductory paragraph.
Response4:Thank you for pointing out this. We have emphasized the research objectives as a separate paragraph at the end of the introduction section.
Comments5:The study period is indicated as 2021–2023, but the text refers to a one-year duration (November 2021 – October 2022). The dates should be made consistent throughout the manuscript.
Response5:Thank you very much for pointing out our mistakes. We have standardized the research period from November 2021 to October 2023.
Comments6:The description of the scale is excessively long and detailed. A more concise summary is recommended, with a reference to the original sources for methodological details.
Response6:Agree. The content of this section has been condensed.
Comments7:The manuscript mentions “Spearman correlation for factor analysis” under statistical analysis, but Spearman is primarily used for correlation, not for factor analysis. This formulation should be corrected.
Response7:Thank you very much for pointing out our mistakes.We have made corrections in the manuscript by deleting "for factor analysis."
Comments8:The absence of randomization and the use of self-reported tools are discussed only in the limitations section; it would be more appropriate to mention them in the Methods section as well.
Response8:Thank you very much for your suggestion. However, after discussion among the authors of this manuscript, we ultimately decided to maintain the original reporting format. We believe that addressing shortcomings in the discussion section is more appropriate. Thank you again for your suggestion.
Comments9:There is an overload of information. The listing of the mean scores for each individual item of the scale makes the text heavy and difficult to read. It is recommended to provide a more concise presentation of the main dimensions, moving the complete list to an appendix.
Response9:Agree. We have retained the description of the overall average in the section "3.3. Scores on the Home Unintentional Injury Environment Scale," but we have removed the specific scores for each dimension. Readers can refer to the table for more information.
Comments10:In the Results section, the presentation of data related to the various types of injuries (falls, external force injuries, burns, poisoning, etc.) is somewhat fragmented, with abrupt shifts from one subsection to another. There is a lack of transition sentences or introductory/concluding remarks that would help guide the reader through the logical progression of the findings. This can make the section harder to follow and may obscure the connections between different injury types.
Consider adding brief transitional or introductory sentences between the subsections (e.g., “Among the different types of injuries, falls were the most frequent, followed by…”). This would improve the narrative flow and enhance overall readability.
Response10:Thank you very much for your guidance! We have re-edited that section according to your suggestion, which not only reflects the internal logic among the data but also enhances the readability of the language.
Comments11:For some injury categories (e.g., poisoning), the number of cases is very low, resulting in odds ratios with wide confidence intervals and non-significant p-values. It is important to highlight this limitation directly in the Results section, not only in the Discussion.
Response11:Thank you very much for your guidance! We have added the sentence at the end of the section “3.4. Analysis of Risk Factors for Unintentional Injuries in the Home”: “It is worth noting that for some injury categories (e.g., poisoning), the number of cases is very low, resulting in odds ratios with wide confidence intervals and non-significant p-values.”
Comments12:The length of the Discussion section is excessive; it could be more impactful if condensed. It is recommended to eliminate repetitions and synthesize the content.
Response12:Thank you very much for your reminder! Since we have already removed the duplicated data between the results and discussion sections in this version of the manuscript, we did not remove the content of the discussion section again in order to discuss the research findings in more detail.
Comments13:Some paragraphs, such as the one addressing injuries caused by household animals, are overly descriptive. The text focuses on general details about the presence of animals in the home, without fully integrating the study’s findings or clearly referencing scientific literature. This weakens the argumentative impact and scientific coherence of the discussion. These parts should be reformulated in a more concise and analytical way, including: the specific results obtained from the study regarding this category of injuries; comparative data from other studies (if available); practical or preventive implications derived from the findings.
Response13:Thank you very much for your suggestion. Based on your revisions, we reviewed studies related to animal injuries, and compared the data with our research findings.We discovered that the incidence rate of animal injuries in our report aligns closely with current studies.Furthermore, we offered several objective recommendations for prevention based on the literature.
Comments14:Regarding the study limitations, while the identified points are appropriate, a more explicit comment should be added about the absence of multivariate analysis to account for potential confounders (e.g., educational level, type of housing).
Response14:Thank you for your guidance. We have incorporated this point into the first limitation.
Comments15:Additionally, it would be advisable to include a recommendation for educational campaigns and policy changes (e.g., incentives for adopting household safety measures), in order to provide more actionable and policy-relevant conclusions.
Response15:Thank you very much for your suggestion. In the conclusion section, we offered framework-based recommendations for preventing accidental injuries to children, focusing on the family, school, and government aspects.
Comments16:The phrase “ongoing attention” is vague. It is recommended to rephrase it in more practical terms, such as: “should be addressed through targeted prevention policies and home safety interventions.”
Response16:Thank you for your guidance. We have changed “ongoing attention” to “continuous attention”.
Reviewer 2 Report
Comments and Suggestions for Authors
Dear authors,
I found this article interesting because understanding the most common causes of accidents in children would allow for the implementation of the necessary preventive measures to avoid them.
The abstract is well-structured and provides information on the main aspects of the research, although I believe the questionnaire used should be included in the methods section.
The introduction addresses the problem of childhood accidents clearly and is based on existing scientific evidence.
The objective is clear and appropriate for addressing the research topic.
The methodology is detailed and would allow for replication of this study, although the type of sampling used should be indicated.
I believe the psychometric properties of the questionnaire used in this study present some values that may influence the results and should be reflected in the discussion and limitations. It is worth highlighting the RMSEA value above 0.5 and the CFI below 0.9.
The results are presented clearly and in detail with tables that facilitate understanding.
Table 3 shows that some items have a standard deviation greater than 2 or less than 0.5. These results could indicate duplication or item interpretation problems, or identical responses from the majority of participants, hindering their discriminatory capacity. These results should be analyzed in the discussion to see if they might suggest eliminating any item from the questionnaire.
The discussion addresses the results in detail, comparing them with existing evidence, although I believe the previously mentioned aspects related to questionnaire validation or item standard deviation should be addressed.
The conclusions adequately address the objectives based on the results obtained.
The references are mostly current, allowing for an adequate approach to the topic of this research.
Kind regards.
Author Response
Dear professor:
I am writing to express my sincere gratitude for the invaluable feedback provided by you on my manuscript titled “The Hidden Danger of Unintentional Child Injuries in Urban Domestic Environment:Considering Unintentional Injuries from Another Angle .” I appreciate the time and effort you have dedicated to evaluating my work, and your insightful comments have significantly enhanced the quality of my submission.
In response to your suggestions, I have made several revisions to improve the clarity and rigor of my manuscript. Specifically, I have addressed the following points:
1. I believe the psychometric properties of the questionnaire used in this study present some values that may influence the results and should be reflected in the discussion and limitations. It is worth highlighting the RMSEA value above 0.5 and the CFI below 0.9.
Response: Thank you for your reminder. In the revised manuscript, we removed the detailed introduction to the scale and discussed this limitation in detail in the discussion.
2. Table 3 shows that some items have a standard deviation greater than 2 or less than 0.5. These results could indicate duplication or item interpretation problems, or identical responses from the majority of participants, hindering their discriminatory capacity. These results should be analyzed in the discussion to see if they might suggest eliminating any item from the questionnaire.
Response: Thank you for your reminder. This point has been added to the discussion.
3. The discussion addresses the results in detail, comparing them with existing evidence, although I believe the previously mentioned aspects related to questionnaire validation or item standard deviation should be addressed.
Response: Thank you for your reminder. This point has been added to the discussion.
Reviewer 3 Report
Comments and Suggestions for Authors
This article, which deals with accidents that occur in the home environment, is very necessary. Because it is very valuable in explaining the factors that cause these accidents and has suggestions for precautions.but a few changes are needed
1.First of all, what is the hypothesis or research question of the study? Please add this after the introduction.
2.There are many deficiencies in the method section. First of all, state that the study is a field study. Also, it would be more accurate if you organized it by creating subheadings under the method. There is a very complicated order that does not create clarity. For example, 1. participant, 2. participant selection, 3. criteria for inclusion in the study, 4. tools used in the study.....
3. How did you calculate the number of participants? Did you use G power or another power analysis... please do this calculation if you haven't done so.Also, how was the sample selected? Which technique was used? Was it simple random or was a special sample selected?
4.tools used in the research; "questionnaire form, scale".... please give under these subheadings
5.After Statistical Analysis, write the title of the ethical dimension of the study and state that you have complied with the Helsinki Declaration in this section. Explain your participant consents. Explain where you received your ethics committee approval and state its number and date. Have you received permission for the use of the scale? Also, state that you have received permission from the institutions where you collected the data.
6.For the findings section, please compare the scale means in terms of demographic data and present Table 1.When you find significance between these averages, add the Cohen's and CIS values to the table. It is important to see how significant the significance is.
7.Consider the findings I suggested for Table 1 in the discussion section. Remember, I am sure that important results will emerge.
Author Response
Dear professor:
I am writing to express my sincere gratitude for the invaluable feedback provided by you on my manuscript titled “The Hidden Danger of Unintentional Child Injuries in Urban Domestic Environment:Considering Unintentional Injuries from Another Angle .” I appreciate the time and effort you have dedicated to evaluating my work, and your insightful comments have significantly enhanced the quality of my submission.
In response to your suggestions, I have made several revisions to improve the clarity and rigor of my manuscript. Specifically, I have addressed the following points:
- First of all, what is the hypothesis or research question of the study? Please add this after the introduction.
Response:Thank you very much for your suggestion. We have added the research hypothesis into the second-to-last paragraph of the introduction based on your comments.
- There are many deficiencies in the method section. First of all, state that the study is a field study. Also, it would be more accurate if you organized it by creating subheadings under the method. There is a very complicated order that does not create clarity. For example, 1. participant, 2. participant selection, 3. criteria for inclusion in the study, 4. tools used in the study.....
Response:Thank you for your valuable suggestions for revisions. We have revised the methods section of the study according to your suggestions.
- How did you calculate the number of participants? Did you use G power or another power analysis... please do this calculation if you haven't done so.Also, how was the sample selected? Which technique was used? Was it simple random or was a special sample selected?
Response:Thank you for your suggestion. The calculation and selection of the sample size have been supplemented in the "Participants Selection" section.
- tools used in the research; "questionnaire form, scale".... please give under these subheadings
Response:Agree. A subheading has been added before the section introducing the scale.
- After Statistical Analysis, write the title of the ethical dimension of the study and state that you have complied with the Helsinki Declaration in this section. Explain your participant consents. Explain where you received your ethics committee approval and state its number and date. Have you received permission for the use of the scale? Also, state that you have received permission from the institutions where you collected the data.
Response:Thank you for your suggestion. We have added ethical-related content as per your suggestion.
6.For the findings section, please compare the scale means in terms of demographic data and present Table 1.When you find significance between these averages, add the Cohen's and CIS values to the table. It is important to see how significant the significance is.
Response:Thank you very much for your suggestion. Since demographic characteristics are not the focus of our study, in order to avoid deviating from the topic, we do not emphasize this part in the manuscript. However, we strongly agree with your viewpoint, and we will emphasize this in reports on other topics.
7.Consider the findings I suggested for Table 1 in the discussion section. Remember, I am sure that important results will emerge.
Response:Thank you very much for your guidance! We have already provided an answer to this question in our response to the sixth question above. We also believe that this will be discussed in detail in the next manuscript.
Reviewer 4 Report
Comments and Suggestions for Authors
The article presents a topic of great interest. The research also includes an important sample, taking into account the ethical considerations that should guide all research. However, the following are some issues that could be improved.
Firstly, the theoretical framework (introduction) is quickly resolved. It is recommended to take an international look at the subject, taking into account what is stated in the world reports on child injury prevention produced by the World Health Organisation in collaboration with Unicef, among others.
It would be advisable to provide more information about the context in which the research takes place, since potential readers of the article do not necessarily need to know the characteristics of the country/city where the research is taking place. In this sense, it would be interesting to specify how many children are usually had per couple and the average age at which they are had, for example.
Both the instrument and the analysis of the results are adequate. However, the discussion of the results can be improved by comparing the results with data from other studies. This discussion is not observed in the current text. In addition, the conclusions are very brief. It is recommended that they be expanded by proposing at the end proposals for action and prevention measures that should not be left solely in the hands of families.
Author Response
Dear professor:
I am writing to express my sincere gratitude for the invaluable feedback provided by you on my manuscript titled “The Hidden Danger of Unintentional Child Injuries in Urban Domestic Environment:Considering Unintentional Injuries from Another Angle .” I appreciate the time and effort you have dedicated to evaluating my work, and your insightful comments have significantly enhanced the quality of my submission.
Based on your suggestions, we have added some content in the relevant sections of the manuscript. We hope that the modifications we made meet your expectations for this manuscript. Thank you once again for your invaluable guidance!
Round 2
Reviewer 2 Report
Comments and Suggestions for Authors
Dear Authors,
I consider that the manuscript has been sufficiently improved and I have no additional comments or suggestions.
Kind regards.
Reviewer 3 Report
Comments and Suggestions for Authors
accepted...good work...
Reviewer 4 Report
Comments and Suggestions for Authors
The authors have improved the article and have responded to some of the recommendations made, and I believe that it meets the requirements for publication.